# Levels of Vitamin D and Expression of the Vitamin D Receptor in Relation to Breast Cancer Risk and Survival

**DOI:** 10.3390/nu14163353

**Published:** 2022-08-16

**Authors:** Linnea Huss, Salma Tunå Butt, Signe Borgquist, Karin Elebro, Malte Sandsveden, Jonas Manjer, Ann Rosendahl

**Affiliations:** 1Department of Surgery, Helsingborg Hospital, 251 87 Helsingborg, Sweden; 2Department of Clinical Sciences, Lund University, 221 00 Lund, Sweden; 3Department of Surgery, Skåne University Hospital, 205 01 Malmö, Sweden; 4Department of Clinical Sciences, Division of Oncology and Pathology, Lund University, Skåne University Hospital, 221 00 Lund, Sweden; 5Department of Oncology, Aarhus University Hospital, Aarhus University, 8200 Aarhus, Denmark; 6Department of Plastic Surgery, Skåne University Hospital, 205 01 Malmö, Sweden

**Keywords:** vitamin D, vitamin D receptor, breast cancer, incidence, survival

## Abstract

Previous research suggests associations between low systemic levels of vitamin D and poor breast cancer prognosis and between expression of the vitamin D receptor (VDR) in breast cancers and survival. This study aimed to study associations between pre-diagnostic systemic levels of vitamin D and expression of VDR in subsequent breast tumors, and interactions between vitamin D and VDR on breast cancer mortality. Systemic vitamin D levels were measured in women within the Malmö Diet and Cancer Study. The expression of VDR was evaluated immunohistochemically in a tissue microarray of subsequent breast cancers. Statistical analyses followed. Women with high levels of vitamin D had a smaller proportion of VDR negative breast tumors compared to women with low levels of vitamin D (odds ratio: 0.68; 95% confidence interval: 0.41–1.13). Vitamin D levels were not found to modify the association between low VDR expression and high breast cancer mortality. To conclude, there was no statistical evidence for an association between pre-diagnostic levels of vitamin D and the expression of VDRs in breast cancer, nor did vitamin D levels influence the association between VDR expression and breast cancer mortality. Further studies are needed in order to establish the effects of vitamin D on breast cancer.

## 1. Introduction

Over the past few years, there has been an increasing interest in studying the potential associations between vitamin D and breast cancer. Results from studies on vitamin D and breast cancer risk have been inconclusive, although several recent studies have reported such an association to be unlikely [1,2,3]. In contrast, several researchers suggest an association between low systemic levels of vitamin D and a worse breast cancer prognosis, with regard to breast cancer-specific or overall mortality [4,5]. A recent epidemiological study also suggests that vitamin D levels are particularly important in the progression of estrogen receptor (ER) positive tumors where high vitamin D levels were associated with a lower risk of recurrence of ER-positive diseases [6].

Previous studies in our research group performed within the Malmö Diet and Cancer Study (MDCS) suggest no association between pre-diagnostic levels of vitamin D and breast cancer risk [7]. However, low levels of vitamin D were associated with a relatively high risk of breast cancer with prognostically unfavorable tumor characteristics [8]. Furthermore, there was a u-shaped association between pre-diagnostic levels of vitamin D and breast cancer-specific survival, with a better prognosis among women within the middle tertile of vitamin D levels [9]. Active vitamin D exerts its functions through binding to the vitamin D receptor (VDR), a nuclear receptor with regulatory effects on gene expression [10,11]. Another study within the MDCS cohort showed that patients with invasive tumors expressing a high fraction of nuclear VDR had a relatively low breast cancer-specific mortality, i.e., a good survival, and were associated with favorable tumor characteristics, such as ER expression [12].

A possible explanation for the reported association between pre-diagnostic vitamin D levels and breast cancer mortality may be that the levels of vitamin D affect the expression of VDR in subsequent breast tumors. Moreover, pre-diagnostic vitamin D levels and VDR interaction may influence breast cancer prognosis. The primary aim of the present study was to analyze if pre-diagnostic systemic levels of vitamin D were associated with VDR expression in subsequent breast tumors, as we have found no previous study investigating this relationship. Secondly, this study aimed to investigate whether an association found between VDR expression and breast cancer mortality was influenced by pre-diagnostic systemic levels of vitamin D.

## 2. Materials and Methods

### 2.1. The Malmö Diet and Cancer Study

During 1991–1996, all women in Malmö, Sweden’s third largest city, born between 1923 and 1950, were invited to take part in a prospective, population-based cohort: The Malmö Diet and Cancer Study (MDCS). Out of eligible participants, 43% of all women (*n* = 17,035) completed baseline examinations; anthropometric measurements, blood sampling, and a self-administered questionnaire investigating lifestyle, reproductive factors, previous diseases, and treatments [13].

### 2.2. Study Population

All women within the MDCS were followed using record linkage with the Swedish cancer registry up until 31 December 2010. Until 31 December 2006, 764 women were diagnosed with breast cancer, and in a previous case-control study they were matched with 764 observations from women with no breast cancer [7]. Due to incidence density matching, there were originally 718 control subjects in the analysis. Within the control group, 18 women were diagnosed with breast cancer until the end of follow-up time for breast cancer diagnosis on 31 December 2010. One woman from the breast cancer case group of 2007 was later concluded misdiagnosed and the control group finally included 701 individuals (Figure 1).

Along with the 18 tumors diagnosed in the control group another 237 women in the MDCS were diagnosed with breast cancer until 31 December 2010, to a total of 1018 women with breast cancer. Out of these tumors, 68 were not invasive and excluded from the present analysis. Another 17 tumors were bilateral at diagnosis and excluded due to difficulties in pairing pathological information. The tissue microarray (TMA) was primarily constructed in order to evaluate survival in relation to the expression of different proteins in tumors. Due to the expected poor prognosis, all tumors with distant metastases at diagnosis (*n* = 14) were excluded. Neoadjuvant chemotherapy prior to surgery might affect the pathology of the tumor and lead to exclusion of four women. One woman was diagnosed with breast cancer at death, one died before surgery, and one refused treatment for four years; these three women were also excluded. In total, 912 women with invasive breast cancer were included in the study population (Figure 1).

### 2.3. Levels of Vitamin D

High-pressure liquid chromatography was used to analyze serum 25-hydroxyvitamin D [25(OH)D] levels, within the previously reported case-control study [7]. Levels of vitamin D were available for 691 of the controls and 677 of the cases in the present study population, whereof 497 also had available nuclear VDR expression. Given seasonal variations in systemic vitamin D levels, all cases and controls with available serum [25(OH)D]-levels were categorized into tertiles within each month of serum sampling, and levels are referred to as low (1st tertile), medium (2nd tertile), or high (3rd tertile) in the present study. After multiple imputations (see below), a new rank of vitamin D levels was used to divide levels into tertiles before statistical analyses.

### 2.4. Clinical Information

Information about lifestyle and reproductive factors was retrieved from the questionnaire included in the MDCS baseline examination. Information on the type of surgery, axillary surgery, and planned adjuvant treatment (as suggested by a multidisciplinary treatment conference following surgery) was retrieved from medical records and clinical notes.

Medical records and pathological reports were used to retrieve information on the size of the tumor and lymph node status. Women included in this study had been diagnosed with breast cancer between 9 December 1991 and 31 December 2010 and the histopathological reports have changed considerably during this time. Hence, information on histological grade, hormone receptor status, proliferation index factor (Ki67), and human epidermal growth factor receptor 2 (HER2) status has been retrieved from different sources during different time periods, as described previously [14,15,16].

Estrogen receptor (ER) and progesterone receptor (PR) status were classified as positive when a fraction exceeding 10% was reported. HER2 status was defined by in situ hybridizations (ISH) when available. When immunohistochemistry was used, a reported value of 3+ was considered positive, 0 or 1+ was negative, and 2+ was defined as missing if ISH could not be used to confirm the result. On the evaluation of Ki67 expression, it was noted that the distribution differed considerably between different periods of diagnosis. Expression of the Ki67 proliferation marker was therefore set as high, intermediate, or low based on tertiles during different time periods (1991–2004, 2005–2007, 2008–2010).

In the South Swedish Health Care Region, molecular subtypes are used for prognostic evaluation [17]. According to the local guidelines a Luminal A-like tumor is ER-positive, HER2-negative, and either, a: histological grade 1 or, b: histological grade 2 and low Ki67 or, c: histological grade 2, intermediate Ki67 and positive PR status. Luminal B-like tumors are ER-positive and HER2-negative and either, a: histological grade 3 or, b: histological grade 2 and high Ki67 or, c: histological grade 2, intermediate Ki67 and negative PR. HER2-positive tumors are all categorized as HER2-positive, regardless of grade or expression of ER, PR, and Ki67. Triple-negative tumors are ER-negative, PR-negative, and HER2-negative.

### 2.5. TMA and VDR Assessment

The TMA had been constructed previously and included breast tumors diagnosed before 31 December 2010. Each of the 718 tumors contributed with two 1-mm cores to the recipient block [18]. The expression of tumor-specific nuclear VDR was evaluated in the TMA by immunohistochemistry using the mouse monoclonal anti-VDR (D-6) antibody (sc-13133, Santa Cruz Biotechnology) as previously reported [12]. In this study, VDR expression was determined based on the fraction of positively stained nuclei in tumor cells and defined as negative if ≤10% of tumor cells had stained nuclei and positive if >10% of tumor cells were stained. Nuclear VDR expression was assessable in 678 of the tumors.

### 2.6. Single Nucleotide Polymorphisms (SNP)

A previous study within the MDCS identified ten SNPs that were associated with levels of vitamin D [19]. Minor alleles of rs12239582 and rs2060793 were associated with high levels of vitamin D, while minor alleles of rs705117, rs2282679, rs7041, rs12295888, rs1007392, rs7944926, rs3829251, and rs2302190 were associated with low levels of vitamin D [19]. These SNPs were included in the first multiple imputation model (see below) but were not included as covariates in any of the final statistical analyses. The process of genotyping and quality control of samples has been described previously [19,20].

### 2.7. Endpoint Retrieval

Women in the MDCS were followed using the Swedish Cause of Death Registry. This registry covers all deaths in Sweden and provided information on the date of death as well as the cause of death. In this study, death from breast cancer was defined as breast cancer registered as the underlying cause of death or contributing cause of death. Other women included in the survival analyses were either alive, dead from another cause, or had emigrated. The end of follow-up was the date of death, date of emigration, or 31 December 2016.

### 2.8. Statistical Methods

Cases and controls were compared to observed differences in the distribution of lifestyle and reproductive factors. All factors differing at least three percentage points between cases and controls were potential confounders and included in the following statistical analyses.

To study the risk of breast cancer, a binary logistic regression analysis, yielding odds ratios (OR) and 95% confidence intervals (CI), was used to analyze associations between tertiles of pre-diagnostic systemic vitamin D-levels and odds of invasive breast cancer overall, and odds of either VDR-negative breast cancer or VDR-positive breast cancer. Crude analyses were followed by analyses adjusted for age and year of inclusion in the MDCS, type of occupation, age at first childbirth, exposure to oral contraceptives, hormonal replacement therapy, and alcohol consumption. As a sensitivity analysis, an additional analysis was performed including body mass index (BMI).

The analyses were run first with only cases and controls with complete data on all variables included (complete case analysis), and thereafter including all cases and controls after multiple imputations of missing data. The imputation model is described in detail in Appendix B.

Tertiles of pre-diagnostic serum levels of vitamin D were compared to note differences in breast tumor characteristics and breast cancer treatment. To investigate associations between tertiles of vitamin D levels and breast cancer mortality, a Cox proportional hazards analysis yielding hazard ratios (HR) and 95% CI:s was performed. The HR of breast cancer death was tested with medium levels of vitamin D (2nd tertile) as a reference group. A multivariable-adjusted model was subsequently constructed, where covariates known to affect levels of vitamin D, such as age and season of diagnosis were included, as well as factors known to influence breast cancer prognosis; the size of the tumor, lymph node status, and molecular subtypes.

Another Cox proportional hazard analysis was used to confirm the previously shown association between VDR expression and breast cancer mortality. Following this, all analyses were repeated, stratified by tertiles of vitamin D, in order to examine whether vitamin D levels influence the association between VDR expression and breast cancer mortality [12]. These models were adjusted for the same confounders as the Cox analysis described above.

Kaplan-Meier estimates were plotted to visualize unadjusted survival associations and to confirm the association of proportional hazards.

All statistical analyses were performed using SPSS (IBM Corp. Released 2017. IBM SPSS Statistics for Windows Version 25.0. Armonk, NY, USA: IBM Corp).

## 3. Results

Levels of vitamin D ranged between 18 and 90 nmol/L within the 1st tertile, between 64 and 116 nmol/L within the 2nd tertile, and between 89 and 288 nmol/L within the 3rd tertile. There is an overlap of levels between tertiles, due to the seasonal variation of vitamin D levels, as tertiles were ranged within the month of collected blood samples. After multiple imputations, a pooled analysis of results from the imputation of vitamin D levels showed ranges from 18 to 90 nmol/L within the 1st tertile, 64 to 111 nmol/L within the 2nd tertile, and from 88 to 288 nmol/L within the 3rd tertile of vitamin D.

As a description of the cohort, the distribution of differences in lifestyle factors and levels of vitamin D between cases and controls are presented in Table 1. It was noticed that women without breast cancer more often had medium levels of vitamin D and that tumors that were not represented in the TMA more often belonged to women with low (1st tertile) or high (3rd tertile) vitamin D.

### 3.1. Vitamin D Levels and Breast Cancer Risk

There was no statistical evidence for an association between levels of vitamin D and the risk of breast cancer, although there was a tendency toward a lower risk of invasive breast cancer among women with medium levels of vitamin D, compared to low levels, in both crude (OR: 0.80, 95% CI: 0.61–1.04) and adjusted (OR: 0.79, 95% CI: 0.60–1.03) analyses using data from multiple imputations (Table 2).

### 3.2. Vitamin D Levels and Expression of VDR in Breast Tumors

Compared to low levels of vitamin D, a statistically non-significant association between high levels of vitamin D and a lower risk of VDR negative tumors was found in the crude (OR: 0.67, 95% CI: 0.41–1.10) and the adjusted (OR: 0.68, 95% CI: 0.41–1.13) analysis (Table 2). When tertiles of vitamin D and risk of VDR-positive tumors were analyzed, the results were similar to overall breast cancer risk with a tendency towards a lower OR within the group of medium levels of vitamin D in the crude (OR: 0.81, 95% CI: 0.62–1.07) and the adjusted analysis (OR: 0.80, 95% CI:0.60–1.06). Adding BMI as a confounder in the sensitivity analysis only marginally attenuated the results. For example, there was a tendency toward a lower risk of a VDR-negative tumor, when high levels of vitamin D were compared to low (OR: 0.72, 95% CI: 0.43–1.21).

### 3.3. Distribution of Tumor Characteristics

The distribution of tumor characteristics in relation to vital status is presented in Table 3, which also includes a comparison between all 912 tumors in the study and the 497 tumors on which information on VDR expression and pre-diagnostic vitamin D levels were available. Tumors with complete information were larger compared to all tumors included in the study.

Distributions of tumor characteristics between different tertiles of vitamin D are presented in Appendix A. As there were differences exceeding three percentage points and possible confounding effects of tumor size, lymph node involvement, and molecular subtypes, these variables were included as confounders in adjusted analyses.

Appendix A shows the distribution of treatment factors in relation to the tertiles of Vitamin D. There were differences exceeding three percentage points in the type of surgery, axillary dissection, planned adjuvant endocrine therapy, and radiotherapy when women with low vitamin D levels were compared to women with medium vitamin D levels.

The Appendix A also present how tumor characteristics and treatment factors differ for women with missing information on VDR and/or vitamin D level, compared to women with complete information.

### 3.4. Survival Analyses

The mean follow-up time for breast cancer mortality was 13.1 years with a standard deviation of 5.8.

Differences in breast cancer survival according to the tertile of vitamin D showed that women with medium levels of vitamin D had a survival rate of 88% compared to women with low or high levels of vitamin D with survival rates of 78% and 81%, respectively (*p* = 0.02, Figure 2A).

There was an association between low levels of vitamin D and a relatively higher risk of breast cancer-specific mortality in the crude analysis (HR: 1.64, 95% CI: 1.05–2.56) when imputed data were used. Moreover, an association between high levels of vitamin D and a relatively higher risk of breast cancer death in both crude and adjusted analyses was seen; however, this result was not statistically significant (Table 4).

Table 4 also presents the results from Cox analyses of associations between VDR expression and breast cancer death. In analyses where all women were included, a negative VDR expression was associated with a relatively higher breast cancer mortality in the crude (HR: 2.08, 95% CI: 1.30–3.32) and the adjusted (HR: 1.62, 95% CI: 0.99–2.65) analyses, although adjustments attenuated the results. The repeated analyses stratified for vitamin D levels, showed that regardless of vitamin D levels a negative VDR expression was associated with a relatively higher breast cancer mortality, but the results were only statistically significant for the crude analysis in the low vitamin D group (HR: 2.19, 95% CI: 1.13–4.25).

Kaplan-Meier curves comparing survival for positive vs. negative VDR, stratified by tertiles of vitamin D are shown in Figure 2B. All curves show a better survival for women with a VDR-positive tumor, although a statistical significance was only reached for the group of women with low pre-diagnostic levels of vitamin D.

## 4. Discussion

The results from this study show no statistically significant evidence that pre-diagnostic vitamin D levels are associated with breast cancer risk. Nevertheless, there were indications that women with high levels of vitamin D were less likely to develop a breast tumor with low VDR expression, compared to women with low levels of vitamin D. Low pre-diagnostic levels of vitamin D were associated with an increased breast cancer mortality. Pre-diagnostic levels of vitamin D did not alter the association between VDR expression in a breast tumor and a favorable breast cancer prognosis.

### 4.1. Vitamin D and Expression of VDR in Breast Cancer

Recent research from large cohorts indicates that there is no association between vitamin D levels and risk of breast cancer [2,3]. The results from the present study are congruent with previous results, although there was a tendency towards a relatively low breast cancer risk for women within the second tertile compared to the first.

Women with high pre-diagnostic levels of vitamin D had a lower percentage of VDR negative tumors compared to women with low or medium levels of vitamin D. This, together with results from Cox analyses showing a lower risk of a VDR negative tumor with increasing levels of vitamin D, indicate that vitamin D levels may influence whether or not a breast tumor express VDR, even though these results did not reach statistical significance. VDR negative tumors were less common than VDR positive tumors, and therefore statistical power was limited in the analysis of VDR negative tumors. 

As there are no publications from previous studies on associations between vitamin D levels and VDR expression, it is not possible to compare our results with others. Our results need to be confirmed in a cohort containing a larger number of tumors.

### 4.2. Vitamin D and Breast Cancer Mortality

In this study, we found that women within the lowest and highest tertiles of pre-diagnostic vitamin D levels had a relatively higher breast cancer mortality compared to women within the medium tertile of vitamin D. Similar results have been reported previously by our group [9]. The current analysis used some additional exclusions, which can explain why this analysis did not find all results to be statistically significant as in our previous study. In the present study, all patients with distant metastasis at diagnosis were excluded as well as women who had received neoadjuvant treatment prior to surgery. Fewer women were included in the complete case analysis as they did not have valid values for VDR expression. Following this, we still find it plausible that a woman’s habitual vitamin-D status does have an impact on breast cancer prognosis, as other researchers also conclude [4].

### 4.3. VDR and Breast Cancer Mortality

The present analysis also found an association between low expression of VDR and a relatively higher risk of breast cancer death in the crude analysis, but when adjustments for factors known to be associated with prognosis were made, hazard ratios were attenuated, and confidence intervals widened which weakened the statistical evidence. A previous study by our group, using the same population showed similar but statistically stronger results [12]. This indicates that including missing data by multiple imputations alters results and should be considered. In particular, VDR positivity was associated with favorable tumor characteristics, as others also have noted [21,22]. When tumor characteristics and VDR status were imputed correctly, this association correlated to an extent that the adjustments for such tumor characteristics diminished the statistical significance.

Our previous study showed a stronger statistical association in the molecular subtype group of Luminal B-like tumors, compared to other subtypes [12], and others have found VDR expression to be predictive for women with multifocal breast cancers [23]. Considering such results, it is possible that analyzing VDR could be of value in certain subgroups of breast cancer, and in others, it may not have any predictive potential.

Another aim of this study was to analyze if the association between VDR expression and breast cancer mortality in any way was affected by pre-diagnostic vitamin D levels. When the analysis was stratified on different tertiles of vitamin D, a pattern with a relatively higher risk of breast cancer death for women with VDR-negative tumors could be seen within all the tertiles, with the strongest association within the lowest tertile of vitamin D. The association did not differ between tertiles to an extent that suggested any effect modification by pre-diagnostic vitamin D levels. In order to investigate if vitamin D levels modify the association between VDR status and prognosis, it would probably be more accurate to compare vitamin D levels from blood samples taken at the time of diagnosis, which we, unfortunately, do not have access to.

### 4.4. Methodological Considerations

Vitamin D ranges used in this study are based on tertiles, as this method gives similarly sized groups which are preferred in statistical analyses. Since vitamin D levels vary with seasons in northern latitudes the groups were based on which month the blood sample was taken, and therefore there were overlaps of vitamin D levels between tertiles. There is still an ongoing debate on recommendations for adequate vitamin D levels. Swedish cut-offs are based on recommendations from the Endocrine Society, <25 nmol/L—deficiency, 25–50 nmol/L—insufficiency, >50 nmol/L—sufficient, 75 nmol/L—optimal level, >125 nmol/L—potentially unhealthy and >250 nmol/L—potentially intoxication [24,25]. A review from 2013 suggests generally higher cut-offs, supposedly adapted to countries with more sun access [26]. When vitamin D-levels within tertiles are compared with Swedish cut-offs, deficient and insufficient levels are within the 1st tertile, and women with levels within the 2nd tertile, are considered to have sufficient levels of vitamin D. Almost all women within the 3rd tertile have vitamin D levels considered normal in sunny countries, which is interesting to note, but compared to the Swedish cut-offs 133 of the women have a measured level of above 125 nmol/L, which is considered potentially unhealthy. As this study did not show any results to suggest higher than sufficient vitamin D levels to be favorable, neither for breast cancer risk nor mortality, we believe that vitamin D substitutes should be advised only to women at risk of, or with confirmed deficiency or insufficiency.

In the present study, test results from two different studies on women within one large cohort were combined. As a consequence, values for vitamin D levels were missing for 12% of subjects and VDR expression was missing for 26% of women in the study population. Still, a study population of 497 women with breast cancer remained, but since there is a probable pattern of missingness at random of both levels of vitamin D and VDR expression we choose to perform multiple imputations to also include women with missing values in the statistical analyses. For instance, when a TMA is used for the analysis of protein expression, small tumors contribute with tissue less often than large tumors. Using multiply imputed data on VDR status diminishes this selection bias to some extent since values on VDR status can be imputed on small tumors, taking tumor size into account. Since data on other predicting variables, such as age and BMI at inclusion and age at the time of diagnosis and size of tumor are complete, the multiple imputation model is considered stable and reliable to impute large numbers of missing values.

In the Cox regression analyses, we chose to adjust for known prognostic factors of breast cancer. These factors are also used when patients are evaluated in the post-surgical treatment conference which suggests adjuvant treatment. We decided not to adjust analyses for treatment factors, which could have led to over-adjusting the statistical model. 

There is a known association between low levels of vitamin D and a high BMI [27], and also between overweight and worse breast cancer prognosis [28]. However, there is a possibility that vitamin D is on the causal pathway between overweight and breast cancer mortality and adding BMI to the analysis may attenuate a true association [9]. Exercise is another factor known to influence both vitamin D and breast cancer prognosis, and the same reasoning can be used if values for exercise would be added to the analysis.

Most previous studies investigating associations between levels of vitamin D and breast cancer have not had access to pre-diagnostic levels of vitamin D but have instead measured vitamin D from blood samples drawn at the time of diagnosis [4,29]. Since the breast cancer itself might influence serum levels of vitamin D, we believe that blood samples drawn before diagnosis better reflect an individual’s habitual vitamin D status, which might influence the environment in which a subsequent breast cancer could develop. It has also been shown previously that there is a low intraindividual variation of levels of vitamin D [30,31], but as oral supplementation of vitamin D has increased over the years [32], there is a risk that women in the cohort have taken supplements and their levels of vitamin D have altered. Unfortunately, we have no information either on supplement usage or serum levels after the baseline examination in the MDCS. This is an obvious limitation of the present study, and future studies are needed to establish possible associations between vitamin D supplementation and vitamin D levels at the time of diagnosis with VDR status in breast tumors.

We hypothesize that women with habitually higher levels of vitamin D could develop breast cancers that are more often VDR positive. They might also have a larger proportion of unliganded VDR present, which in a previous study has been found to have tumor proliferative effects [33], and hence, through this pathway, may be associated with a slightly increased risk of breast cancer-associated death compared to women with medium levels of vitamin D. Low levels of vitamin D, on the other hand, may be associated with tumors that are VDR negative to a greater proportion and the association between a worse breast cancer prognosis and low levels of vitamin D may be mediated through a deficiency of protective effects from liganded VDR.

Future studies should be performed in order to identify subgroups of breast cancer patients that could benefit from vitamin D supplementation along with conventional breast cancer treatment. Further studies on associations between tumor characteristics and host factors with breast cancer prognosis are needed.

## 5. Conclusions

The hypothesis that pre-diagnostic vitamin D levels would associate with VDR expression in subsequent breast cancer could not be confirmed with statistical evidence in this study. The association between VDR expression and breast cancer prognosis was not modified by pre-diagnostic levels of vitamin D. It would be preferable to investigate whether vitamin D levels at diagnosis and VDR expression in a tumor coincide to influence breast cancer prognosis.

## Figures and Tables

**Figure 1 nutrients-14-03353-f001:**
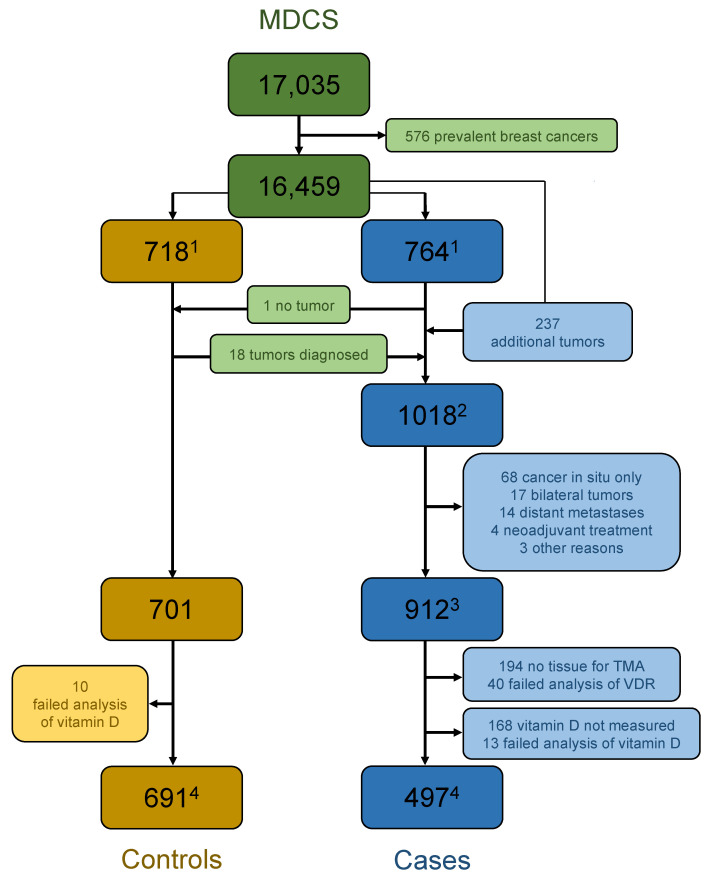
Flowchart of study population, inclusion, and exclusion criteria. ^1^ Cases until end 2006 and controls in study with vitamin D levels measured. ^2^ Cases until end 2010. ^3^ Cases included in TMA. ^4^ Cases and controls with information on vitamin D level and VDR expression.

**Figure 2 nutrients-14-03353-f002:**
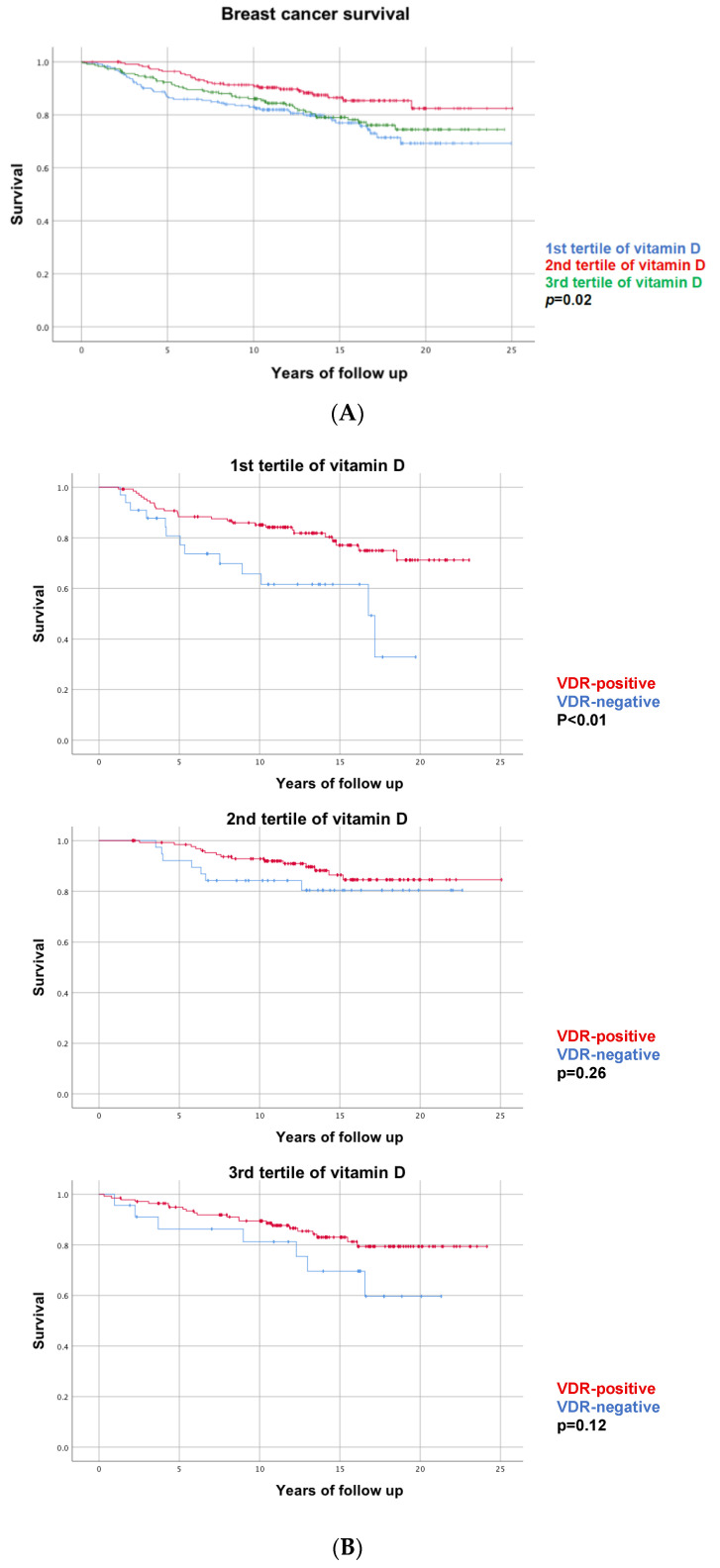
(**A**) Kaplan-Meier curve showing breast cancer survival by tertiles of vitamin D. (**B**) Kaplan-Meier curves showing breast cancer survival by VDR expression stratified by tertiles of vitamin D.

**Table 1 nutrients-14-03353-t001:** Baseline characteristics of breast cancer cases and control subjects.

Factor	Category	Study Population			
		Controls (*n* = 701)		Cases (*n* = 912)	
		Controls (*n* = 691)	Missing ^1^ (*n* = 10)	Cases (*n* = 497)	Missing ^2^ (*n* = 415)
		Column %			
Age (Years)	Mean ± Standard Deviation	57.1 ± 7.3	57.7 ± 7.5	56.7 ± 7.1	56.2 ± 7.2
Year of baseline examination	1991	9.3	0.0	8.9	9.9
	1992	19.4	10.0	20.3	17.6
	1993	20.8	10.0	21.1	18.6
	1994	18.1	40.0	17.7	19.0
	1995	20.8	30.0	20.3	21.9
	1996	11.6	10.0	11.7	13.0
Education	O-level college	68.0	80.0	67.2	67.7
	A-level college	7.7	0.0	7.0	6.5
	University	24.3	20.0	25.8	25.1
Type of occupation	Manual worker	38.2	30.0	32.8	31.1
	Non-manual worker	53.7	50.0	60.6	62.2
	Employer/self-employed	8.0	20.0	5.4	5.5
	Missing	0.1	0.0	1.2	1.2
Age at menarche (years)	<12	4.5	0.0	7.0	7.5
	12–15	68.6	80.0	65.8	69.6
	>15	26.3	20.0	26.4	21.9
	Missing	0.6	0.0	0.8	1.0
Parity	Nullipara	11.1	0.0	11.9	15.9
	1 child	21.6	40.0	18.5	21.4
	2 children	41.5	50.0	45.3	43.6
	3 children or more	23.0	10.0	21.9	16.9
	Missing	2.7	0.0	2.4	2.2
Age at first birth (years)	Nullipara	11.1	0.0	11.9	15.9
	≤20	18.4	10.0	14.5	17.1
	21–24	27.6	0.0	29.4	25.8
	25–29	28.5	40.0	29.2	26.7
	≥30	11.6	50.0	12.7	12.3
	Missing	2.7	0.0	2.4	2.2
Age at menopause (years)	Pre-/Perimenopausal	32.0	30.0	34.8	38.8
	<45	9.1	10.0	8.0	9.2
	45–53	42.4	30.0	41.4	35.2
	>53	15.2	30.0	14.1	14.2
	Missing	1.3	0.0	1.6	2.7
Exposure to oral contraceptives	No	50.9	50.0	46.9	42.4
	Yes	49.1	50.0	53.1	57.3
Exposure to hormonal replacement therapy (HRT)	No (premenopausal)	25.9	20.0	27.0	31.6
	No (postmenopausal)	54.1	50.0	43.7	42.9
	Estrogen only	8.8	20.0	7.2	7.5
	Progesterone only	0.7	0.0	1.2	1.4
	Combined HRT	10.0	10.0	20.7	16.1
Body mass index (kg/m^2^)	<25	51.7	50.0	51.5	51.1
	≥25 < 30	36.3	40.0	32.4	35.9
	≥30	12.0	10.0	16.1	13.0
Alcohol consumption	Nothing last year	11.4	10.0	10.5	6.5
	Something last year	12.3	10.0	9.9	12.8
	Something last month	76.0	80.0	79.5	80.5
Tertile of vitamin D	1st	31.4	0.0	34.0	36.7
	2nd	35.2	0.0	33.0	27.8
	3rd	33.4	0.0	33.0	35.6

Separate missing categories given only if missing ≥ 1%. ^1^ Missing vitamin D level. ^2^ Missing tumor in tissue microarray (*n* = 194), VDR expression (*n* = 40), or vitamin D level (*n* = 181).

**Table 2 nutrients-14-03353-t002:** Levels of vitamin D in relation to risk of breast cancer overall, or risk of VDR neg/pos breast cancer.

		Complete Case Analysis *			All Included **		
	Vitamin D Tertile/Level	Case/ Controls	OR (95% CI)	OR ^1^ (95% CI)	Case/ Controls	OR (95% CI)	OR ^1^ (95% CI)
Invasive breast cancer	1st Low	169/217	1.00 (ref)	1.00 (ref)	317/220	1.00 (ref)	1.00 (ref)
	2nd Medium	164/243	0.81 (0.63–1.06)	0.82 (0.63–1.07)	289/251	0.80 (0.61–1.04)	0.79 (0.60–1.03)
	3rd High	164/231	0.91 (0.70–1.18)	0.91 (0.69–1.19)	306/230	0.92 (0.71–1.20)	0.90 (0.69–1.18)
	Total	497/691			912/701		
VDR-negative breast cancer	1st Low	36/217	1.00 (ref)	1.00 (ref)	66/220	1.00 (ref)	1.00 (ref)
	2nd Medium	35/243	0.87 (0.53–1.43)	0.94 (0.55–1.60)	56/251	0.74 (0.47–1.18)	0.73 (0.46–1.17)
	3rd High	23/231	0.60 (0.35–1.05)	0.64 (0.36–1.16)	47/230	0.67 (0.41–1.10)	0.68 (0.41–1.13)
	Total	94/691			169/701		
VDR-positive breast cancer	1st Low	133/217	1.00 (ref)	1.00 (ref)	251/220	1.00 (ref)	1.00 (ref)
	2nd Medium	129/243	0.87 (0.64–1.17)	0.86 (0.63–1.17)	233/251	0.81 (0.61–1.07)	0.80 (0.60–1.06)
	3rd High	141/231	1.00 (0.74–1.35)	0.97 (0.71–1.32)	259/230	0.99 (0.75–1.30)	0.96 (0.72–1.27)
	Total	403/691			743/701		

* Only cases and controls with no missing values included in analyses. ** Multiple imputations used to include all cases and controls with missing values. ^1^ Adjusted for age at baseline, year of baseline, type of occupation, age at first childbirth, exposure to oral contraceptives, exposure to hormone replacement therapy, and alcohol consumption.

**Table 3 nutrients-14-03353-t003:** Vital status in relation to age at diagnosis and prognostic factors for breast cancer.

Factor		Survival							
		Cases with Complete VDR & Vitamin D				All Cases *			
		Alive *n* = 309	Dead Breast Cancer *n* = 91	Dead Other Cause *n* = 94	Total ** *n* = 497	Alive *n* = 599	Dead Breast Cancer *n* = 154	Dead Other Cause *n* = 156	Total ** *n* = 912
		Column percent Mean (SD) in bold *Missing numbers in (italics)*							
Age at diagnosis		**62.6 (7.2)**	**64.8** **(8.5)**	**68.6** **(7.4)**	**64.1** **(7.8)**	**64.1** **(7.6)**	**66.4** **(9.1)**	**69.7** **(7.8)**	**65.4** **(8.1)**
Season of diagnosis	January–March	24.9	29.7	25.5	26.0	26.0	27.9	26.3	26.4
	April–June	27.8	19.8	18.1	24.7	26.7	20.1	17.9	24.2
	July–September	15.9	19.8	25.5	18.3	18.7	23.4	24.4	20.4
	October–December	31.4	30.8	30.9	31.0	28.5	28.6	31.4	28.9
Size	1–10 mm	27.1	2.2	22.3	21.7	31.2	4.1	25.5	25.7
	11–20 mm	50.7	41.8	43.6	47.6	47.5	41.9	45.1	46.1
	≥21 mm	22.2	56.0	34.0	30.8	21.3	54.1	29.4	28.2
	*Unknown*	*(3)*	*(0)*	*(0)*	*(3)*	*(16)*	*(6)*	*(3)*	*(25)*
Lymph node status	Positive	29.5	60.7	25.3	35.0	24.9	58.7	31.1	32.0
	Negative	70.5	39.3	74.7	65.0	75.1	41.3	68.9	68.0
	*Unknown*	*(24)*	*(2)*	*(7)*	*(34)*	*(57)*	*(11)*	*(24)*	*(93)*
Grade	I	26.8	14.8	27.8	25.1	30.7	10.9	28.3	27.2
	II	52.3	33.0	46.7	47.6	48.7	42.3	44.8	46.9
	III	20.9	52.3	25.6	27.3	20.5	46.7	26.9	25.9
	*Unknown*	*(7)*	*(3)*	*(4)*	*(14)*	*(49)*	*(17)*	*(11)*	*(77)*
Histological type	Ductal	73.0	71.6	70.3	72.0	71.2	68.8	71.9	70.8
	Lobular	20.7	23.9	18.7	21.0	19.0	25.4	17.1	19.8
	Other/mixed	6.3	4.5	11.0	7.0	9.8	5.8	11.0	9.4
	*Unknown*	*(5)*	*(3)*	*(3)*	*(11)*	*(46)*	*(16)*	*(10)*	*(72)*
ER	Negative 0–10%	12.6	18.8	9.3	13.2	10.3	17.5	6.5	10.8
	Positive > 10%	87.4	81.3	90.7	86.8	89.7	82.5	93.5	89.2
	*Unknown*	*(24)*	*(11)*	*(8)*	*(43)*	*(83)*	*(34)*	*(18)*	*(135)*
PgR	Negative 0–10%	46.5	62.3	46.9	49.3	38.1	55.6	42.6	41.6
	Positive > 10%	53.5	37.7	53.1	50.7	61.9	44.4	57.4	58.4
	*Unknown*	*(38)*	*(14)*	*(13)*	*(65)*	*(103)*	*(37)*	*(27)*	*(167)*
HER2 status	Negative	91.1	84.3	94.8	90.6	91.7	85.8	91.9	90.8
	Positive	8.9	15.7	5.2	9.4	8.3	14.2	8.1	9.2
	*Unknown*	*(52)*	*(21)*	*(17)*	*(91)*	*(128)*	*(41)*	*(32)*	*(202)*
KI67 status	Low	44.1	27.2	32.1	38.7	46.2	27.8	34.2	40.8
	Intermediate	31.6	34.6	39.5	33.6	29.3	31.3	36.7	31.0
	High	24.3	38.3	28.4	27.7	24.5	40.9	29.2	28.3
	*Unknown*	*(62)*	*(10)*	*(13)*	*(86)*	*(203)*	*(39)*	*(36)*	*(279)*
Molecular subtypes	Luminal A-like	60.1	33.8	56.9	55.0	60.5	33.0	57.7	55.7
	Luminal B-like	20.6	33.8	29.2	24.5	21.9	35.0	27.9	25.0
	HER2+	9.7	16.2	5.6	10.0	9.4	16.0	9.0	10.4
	Triple-negative	9.7	16.2	8.3	10.5	8.2	16.0	5.4	8.9
	*Unknown*	*(71)*	*(23)*	*(22)*	*(117)*	*(184)*	*(54)*	*(45)*	*(284)*
VDR expression	Negative 0–10%	15.2	29.7	21.3	18.9	14.9	30.6	20.9	18.4
	Positive > 10%	84.8	70.3	78.7	81.1	85.1	69.4	79.1	81.6
	*Unknown*	*(0)*	*(0)*	*(0)*	*(0)*	*(150)*	*(43)*	*(41)*	*(234)*
Vitamin D tertile	1st Low	30.1	41.8	39.4	34.0	31.8	39.8	39.5	34.7
	2nd Medium	37.2	25.3	26.6	33.0	35.1	23.6	27.9	31.6
	3rd High	32.7	33.0	34.0	33.0	33.2	36.6	32.6	33.7
	*Unknown*	*(0)*	*(0)*	*(0)*	*(0)*	*177*	*31*	*27*	*(235)*

Percentages calculated with missing categories excluded. * All cases included when multiple imputations were applied. ** Three emigrated women not included in the table.

**Table 4 nutrients-14-03353-t004:** Vitamin D levels in relation to breast cancer mortality.

		Complete Case Analysis *					All Included **			
Vitamin D Tertile/Level	VDR Expression	Breast Cancer Mortality/ 100,000	Numbers within Group	Dead Breast Cancer	HR (CI 95%)	HR ^1^ (CI 95%)	Numbers within Group	Dead Breast Cancer	HR (CI 95%)	HR ^1^ (CI 95%)
1st Low	All tumors	1849	169	38	**1.72** **(1.03–2.89)**	1.25 (0.67–2.34)	301	59	**1.64** **(1.05–2.56)**	**1.56** **(1.00–2.47)**
2nd Medium		1082	164	23	1.00 (ref)	1.00 (ref)	307	40	1.00 (ref)	1.00 (ref)
3rd High		1414	164	30	1.32 (0.76–2.27)	1.47 (0.79–2.71)	304	56	1.44 (0.91–2.28)	1.41 (0.88–2.25)
All	VDR neg	2507	94	27	**2.02** **(1.29–3.17)**	1.49 (0.83–2.67)	201	54	**2.08** **(1.30–3.32)**	1.62 (0.99–2.65)
	VDR pos	1225	403	64	1.00 (ref)	1.00 (ref)	711	100	1.00 (ref)	1.00 (ref)
1st Low	VDR neg	3542	36	13	**2.35** **(1.20–4.59)**	1.53 (0.53–4.44)	73	22	**2.19** **(1.13–4.25)**	1.60 (0.74–3.45)
	VDR pos	1481	133	25	1.00 (ref)	1.00 (ref)	227	37	1.00 (ref)	1.00 (ref)
2nd Medium	VDR neg	1638	35	7	1.71 (0.70–4.16)	1.95 (0.65–5.84)	68	14	1.97 (0.87–4.47)	1.41 (0.54–3.70)
	VDR pos	942	129	16	1.00 (ref)	1.00 (ref)	240	26	1.00 (ref)	1.00 (ref)
3rd High	VDR neg	2475	23	7	1.89 (0.81–4.41)	1.32 (0.46–3.83)	60	18	2.03 (0.93–4.40)	1.85 (0.81–4.22)
	VDR pos	1251	141	23	1.00 (ref)	1.00 (ref)	244	38	1.00 (ref)	1.00 (ref)

* Only cases and controls with no missing values included in analyses. ** Multiple imputations are used to include all cases with missing values. ^1^ Adjusted for age at and season of diagnosis, the size of the tumor, lymph node status, and molecular subtypes. Results of statistical significance are indicated in bold.

## Data Availability

The data that support the findings of this study are available on request from the corresponding author (L.H.). The data are not publicly available due to Swedish restrictions.

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
