# Peer review of "Levels of Vitamin D and Expression of the Vitamin D Receptor in Relation to Breast Cancer Risk and Survival"

_nutrients, 2022, doi:10.3390/nu14163353_

Round 1
Reviewer 1 Report
The authors investigated the interaction between pre-diagnostic vitamin D levels and the expression of vitamin D receptor in subsequent breast tumors on breast cancer prognosis.
The question is original and the manuscript is well-written. All the sections of the manuscript are well-structured and clear. The limitations of the study are properly discussed. The English language and style require fine/minor spell check.
In order to further improve the manuscript, I would suggest to the authors to be more careful in the discussion and throughout the main text (i.e. in the beginning of the discussion section) in describing and stating the results that do not reach statistical significance.
The Figure 2B is not clear to me, it seems stratified by tertiles of vitamin D (as Figure 2A) and not by VDR expression.
Did the authors investigate if menopausal status and time from blood withdrawal to diagnosis could impact the results? It should be interesting to investigate and discuss.
Reviewer 2 Report
The interest of the scientific community in researching the implications of vitamin D in the prevention and treatment of different diseases, especially oncological ones, is growing and we should support the efforts in this regard. Even if the results did not reach statistical significance i consider an important fact that it is the first one who evaluated the relationship between vitamin D levels and VDR expression among breast cancer patients and their work could be significant for researchers who work in this field. However, i I would like the authors to include an idea of how this research can help clinicians in daily practice regarding vitamin D recommendations for both prevention or treatment – and include further research perspectives based on the group’s expertise. However, a negative aspect is related to the use of English language which require moderate to extensive improvements.
Line 39 – remove „found”
I recommend including in introduction section a small idea about nuclear vitamin D receptor (VDR)
Line 53-54 recommend rephrasing „pre-diagnostic vitamin D levels and VDR 53 may interact regarding breast cancer prognosis” intro „pre-diagnostic vitamin D levels and VDR interaction may influence breast cancer prognosis”
Line 66 – rephrase „questionnaire with questions”
I recommend mentioning in section 2.1 where can we consult the questionnaire
Line 69 – replace „followed” with recruited
Line 91 – add „criteria” at the end of Figure 1 title -> Flowchart of study population, inclusion, and exclusion criteria
In section 2.6 i recommend mentioning which SNPs were associated with higher levels and which with lower levels
In Figure 2 A and B I recommend separating the legend from the graphic
Line 300 – 301 – replace “have already been published by us” with Have been reported previously by our group
Line 301 – replace “which can explain that this “ with “which can explain why this”
Line 302 – remove “that is”
Line 311 – replace “was made” with “were made”
Line 312 – rephrase “A previous study by us”
Line 311-312 – explain “the results were attenuated and weakened statistical evidence”
Line 317 – remove may have
According to the journal guidelines, ‘doi’ is not included in the references. Please check.
Round 2
Reviewer 2 Report
The paper in its actual form can be accepted for publication